# Ferroptosis: Emerging Role in Diseases and Potential Implication of Bioactive Compounds

**DOI:** 10.3390/ijms242417279

**Published:** 2023-12-08

**Authors:** Giuseppe Tancredi Patanè, Stefano Putaggio, Ester Tellone, Davide Barreca, Silvana Ficarra, Carlo Maffei, Antonella Calderaro, Giuseppina Laganà

**Affiliations:** Department of Chemical, Biological, Pharmaceutical and Environmental Sciences, University of Messina, Viale Ferdinando Stagno d’Alcontres 31, 98166 Messina, Italy; giuseppe.patane@studenti.unime.it (G.T.P.); davide.barreca@unime.it (D.B.); silvana.ficarra@unime.it (S.F.); carlo.maffei@studenti.unime.it (C.M.); antocalderaro@gmail.com (A.C.); giuseppinalagana@unime.it (G.L.)

**Keywords:** ferroptosis, iron, lipid peroxidation, glutathione peroxidase, Xc^−^ system, cancer, natural antioxidants

## Abstract

Ferroptosis is a form of cell death that is distinguished from other types of death for its peculiar characteristics of death regulated by iron accumulation, increase in ROS, and lipid peroxidation. In the past few years, experimental evidence has correlated ferroptosis with various pathological processes including neurodegenerative and cardiovascular diseases. Ferroptosis also is involved in several types of cancer because it has been shown to induce tumor cell death. In particular, the pharmacological induction of ferroptosis, contributing to the inhibition of the proliferative process, provides new ideas for the pharmacological treatment of cancer. Emerging evidence suggests that certain mechanisms including the Xc^−^ system, GPx4, and iron chelators play a key role in the regulation of ferroptosis and can be used to block the progression of many diseases. This review summarizes current knowledge on the mechanism of ferroptosis and the latest advances in its multiple regulatory pathways, underlining ferroptosis’ involvement in the diseases. Finally, we focused on several types of ferroptosis inducers and inhibitors, evaluating their impact on the cell death principal targets to provide new perspectives in the treatment of the diseases and a potential pharmacological development of new clinical therapies.

## 1. Introduction

Cell death is a physiological end-of-life process generally caused by irreversible damage. In general, two distinct types of cell death can be distinguished: accidental cell death and regulated cell death. Regulated cell death relies on specialized molecular mechanisms and can be delayed or accelerated. A normal cell is limited to a defined range of functions established by its genetic program. When the cell is no longer able to respond to physiological demands, an irreversible process of cell death begins. There are different types of cell death, characterized by the different pathways involved: the best known are necrosis and apoptosis. The first caused by physical or chemical damage of an external nature; the second, called cell suicide, occurs during development and morphogenesis. Among other types of cell death, autophagy is a catabolic process of the selective degradation of damaged organelles and macromolecules by lysosomes, while ferroptosis is a type of iron-dependent cell death caused by oxidative-mediated phospholipid damage, which has been recently discovered [1].

Study of the cell death process is important because it represents one of the potential pharmacological points of intervention to avoid the onset of some pathological events and to facilitate the development of therapeutic strategies. In this context, evidence suggests ferroptosis as one of the main cell death mechanisms promoting degenerative diseases, but contextually this process seems to be a new hope for many cancer therapies. In the past decade, the great bioactive potential of natural compounds against ferroptosis has been affirmed. In this review, the mechanism of ferroptosis and its involvement in some of the main diseases affecting human health will be highlighted by evaluating the role of some inducers and inhibitors in the cell death process.

## 2. Ferroptosis: An Overview

Ferroptosis is an iron-dependent programmed cell death that was first identified by Dr. Brent R. Stockwell and described by Dixon in 2012. The main signal of the process is iron accumulation, an increase in oxygen reactive species (ROS), and lipid peroxidation (see Table 1) [2,3].

From a morphological point of view, ferroptosis is mainly manifested by the narrowing of mitochondria (shrinkage), an increase in membrane density, and a reduction in mitochondrial cristae, followed by rupture of the outer membrane (see Table 1) [2,4]. Ferroptosis biological processes mainly include altered iron homeostasis leading to intracellular accumulation of Fe^2+^, decrease in reduced glutathione (GSH) and consequent decreased activity of Glutathione peroxidase 4 (GPx4), increase in radical species, and lipid peroxidation. Interestingly, oxidative stress and lipid peroxidation, two of the main features of ferroptosis, are also two key processes in cancer immunotherapy. This is an emerging treatment that functions to modulate the activity of immune cells, activating them against the processes of carcinogenesis; thus, this therapy does not act directly on the tumor but acts in modulating the immune system. Studies have shown that T lymphocytes activated during therapy increase lipid peroxidation, thus leading to oxidative stress, a mechanism underlying ferroptosis. Wang et al. in their study saw how CD8+ T cells, involved in neutralizing tumor cells, release proinflammatory cytokines, such as tumor necrosis factor (TNF) and gamma interferon. These molecules inhibit the expression of Solute Carrier Family 7 Member 11 (SLC7A11) and Solute Carrier Family 3 Member 2 (SLC3A2), two subunits of the xc system, and further direct the target (tumor) cell toward ferroptosis. The study carried out by the scholars thus provides insight into how ferroptosis could be induced within the body as a defense tool against the processes of carcinogenesis [2,5,6] Ferroptosis may be closely linked to the pathogenesis of various conditions including aging, neurodegenerative diseases, cardiovascular diseases, metabolic diseases, and autoimmune diseases [7,8,9,10]. For example, in Parkinson’s disease, iron accumulation in dopaminergic neurons can lead to ferroptosis and neuronal death. Other neurodegenerative diseases in which a correlation with ferroptosis has been observed are Alzheimer’s disease (AD), Huntington’s disease (HD), and Amyotrophic lateral sclerosis (ALS), which are pathologies that we will discuss in more detail in the following paragraphs. In addition, it has been suggested that ferroptosis may play a role in ischemia/reperfusion injury, where the accumulation of iron and subsequent lipid peroxidation contribute to tissue damage [11]. Ferroptosis can also be used against cancer cells; in fact, one of the main and more dangerous features of cancer cells is escape from cell death. Several studies have demonstrated that ferroptosis induction can inhibit the proliferation of tumor cells, providing new promising therapeutic strategies for cancer treatment [12,13,14,15,16]. Induction of ferroptosis could be useful both in determining the death of cells resistant to classic anticancer treatments and as an adjunct to classic therapies. Indeed, it has been seen in a study by Roh J.L. et al. that ferroptotic inducers, such as cisplatin, can adjunct the action of classic anticancer drugs by inhibiting tumor cell proliferation in mouse models [17]. In addition, in a work conducted by Chen X. et al. it was seen that the induction of ferroptosis for cancer therapy is closely related to the type of cancer we consider. In fact, some cancers such as melanoma, breast cancer, etc., which are characterized by iron accumulation, elevated fatty acid synthesis, increased autophagic processes, and the EMT mechanism (epithelial-mesenchymal transition, leads epithelial cells to conversion to mesenchymal cells), are more susceptible to ferroptosis, making possible the application of this strategy [18]. In summary, ferroptosis is a complex process characterized by multiple factors including iron accumulation, oxidative stress, and inflammation, and it is involved in the genesis of a wide spectrum of serious diseases that are still not effectively controlled and eradicated. Study of the molecular mechanisms and signaling pathways of ferroptosis, although very complex, can contribute to opening new therapeutic opportunities. It has already been shown that ferroptosis can be pharmacologically inhibited with iron chelators, such as deferoxamine, and lipid peroxidation inhibitors such as ferrostatin [19]. Ferroptosis can instead be induced by mutations of RAS, which may cause iron accumulation following activation of transferrin receptor 1 (TfR1) and suppression of iron storage proteins [19,20].

## 3. Mechanism of Ferroptosis

Although the complexities of the molecular processes involved in Ferroptosis are still not enough described and fully understood, the main biological pathways and regulatory factors are iron homeostasis, lipid metabolism, and antioxidant defense systems (see Figure 1) [3].

### 3.1. Iron Homeostasis

Iron is an essential trace metal in the body, where it may exist in two oxidation states: ferrous iron (Fe^2+^) and ferric iron (Fe^3+^). Iron may take or lose electrons; in a Fenton reaction, ferrous iron gives an electron to hydrogen peroxide (H_2_O_2_) to form hydroxyl radical (OH^∙^), a high reactive free radical [21]. The majority of iron ions are bound to iron-binding proteins and biomolecules that exploit its reactivity, limiting damage. Iron is bound to the active site of numerous enzymes, including Catalase, Aconitase, Succinate dehydrogenase, Cytochrome P450, Peroxidases, etc. [22]. Iron also plays a key role in several biochemical processes including deoxyribonucleic acid (DNA) synthesis, electron transport for ATP synthesis, and oxygen transport to tissues [23]. Despite its biochemical use, a disorder in the distribution and content of the body free iron could disrupt cell survival itself. In the Fenton reaction Fe^2+^ + HOOH → Fe^3+^ + OH^−^ + OH^∙^, iron salts react with peroxides to give highly harmful hydroxy radicals; this means that the intracellular free iron levels must be kept in a very narrow range [24]. Normally, excess Fe^2+^ ions are bound to the highly conserved iron-binding protein ferritin, which is an intracellular protein with ferroxidase activity that stores ferrous iron (Fe^2+^), converting it to ferric (Fe^3+^) forms [22]. When the body needs iron, ferritin releases the metal that reaches the bloodstream via ferroportin; ferroportin and divalent metal transporter 1 (DMT1) are the two main iron transmembrane transporters: an extracellular exporter and an intracellular importer, respectively. In the blood, iron circulates bound to transferrin (Tf), which is a glycoprotein consisting of a single polypeptide chain that possesses two binding sites for the ferric ion (Fe^3+^). Tf delivers Fe^3+^ ions to various tissues after binding to its specific receptor (TfR) on the cell surface. The importance of iron in many biochemical processes and its high hazard linked to the Fenton reaction make a constant balance between metal uptake, transport, storage, and utilization to maintain iron homeostasis extremely important [25]. The main proteins regulating iron homeostasis are hepcidin, which regulates the flow of iron from cells into the systemic circulation, and iron regulatory proteins (IRP1 and 2) that mediate regulation of intracellular iron homeostasis [26]. Dysregulation of IRP1 and/or IRP2 can lead to iron level changes and ferroptosis regulation [27]. Ferroptosis can be prevented or possibly also induced by iron chelating agents including deferoxamine, deferiprone, and deferasirox which, by chelating to iron, reduce its availability [2,27]. Generally, all regulators of iron metabolism may be involved in the regulation of ferroptosis as well as ferritinophagy, a selective form of autophagy which may contribute to the onset of ferroptosis through ferritin degradation [28]. Specifically, ferritinophagy is mediated by nuclear receptor coactivator 4 (NCOA4), a selective cargo receptor which binds microtubule-associated proteins to the developing autophagosome membrane [29]. NCOA4 levels, in turn, are modulated by cellular iron levels in a way that high iron pressure decreases NCOA4-E3 ubiquitin–protein ligase HERC2 binding, avoiding elevated NCOA4 degradation through the ubiquitin proteasome system when the cellular iron levels are high [30]. Ferritinophagy via NCOA4 is required for erythropoiesis and is regulated by iron-dependent HERC2-mediated proteolysis [31]. These findings highlight NCOA4 and HERC2 as further potential regulators of ferroptosis. An increase in NCOA4 leads to ferritin degradation and intracellular ferrous iron increase, triggering ferroptosis, while the knockdown of NCOA4 levels blocks ferritin degradation, also suppressing erastin-induced ferroptosis in pancreatic cells [32]. Loss of coatomer subunit zeta-1 (COPZ1) induces NCOA4-mediated autophagy and ferroptosis in glioblastoma cell lines [33].

### 3.2. Lipid Metabolism

Unlike many other programmed cell deaths, ferroptosis is characterized by high levels of lipid peroxidation. The cell membrane is principally constituted of phosphatidylcholine, phosphatidylethanolamine, phosphatidylserine, and sphingomyelin, which together account for more than half of the total lipid. In addition, phospholipids, cholesterol, and glycoprotein are found; among them, polyunsaturated fatty acids (PUFAs) are the most susceptible to peroxidation. Normally, long-chain-PUFAs are converted in PUFA-CoA by Acyl-CoA synthase long-chain family member 4 (ACSL4) to facilitate their entry into the phospholipid bilayer [34]. In the peroxidation process, membrane-bound lipids containing PUFAs are mainly attacked because of the existence of a diallyl matrix that makes them highly susceptible to free radicals and lipoxygenases (LOXs). LOXs are iron-containing enzymes which catalyze the enzymatical lipid peroxidation chain, one of the main characteristics of ferroptosis. LOXs’ action oxidizes PUFAs, generating the corresponding hydroperoxide derivatives which, in turn, generate more aldehydes including 4-hydroxynonenal and malondialdehyde, triggering a dangerous chain reaction on the phospholipid bilayer that ultimately destructs the membrane’s integrity [7]. Another mechanism of lipid peroxidation involved in ferroptosis occurs by spontaneous non-enzymatic autoxidation. In this case, the oxidation trigger is an abnormal accumulation of free ferrous iron that through the Fenton reaction reacts with hydrogen peroxide-generating ferric iron and hydroxyl radicals [35]. The latter, highly unstable, damages the membrane by initiating the process of lipid peroxidation by extracting hydrogen from lipids containing carbon–carbon double bonds, especially from PUFAs [36]. Membrane oxidative damage is the final step of ferroptosis [37].

### 3.3. Antioxidant Defense Systems

The human body is equipped with a variety of antioxidant agents to counteract the dangerous action of oxidants and free radicals. In general, these can be divided in enzymatic and non-enzymatic antioxidants. The major enzymatic antioxidants are superoxide dismutase (SOD), catalase, and GPx. Among the intracellular antioxidant elements, the system Xc^−^ plays an important role. The system Xc^−^ is a glutamate–cystine antiport that takes cystine from outside the cell and excretes glutamate in a 1:1 ratio [38,39]. The system Xc⁻ is part of the heterodimeric amino acid transport family; the system widely distributed on cell membranes is composed of two subunits: a light chain SLC7A11, responsible for amino acid exchange, linked by a disulfide bridge to a heavy chain SLC3A2 which is a chaperone [38,40]. Inside the cell, cystine reductase reduces imported cystine to cysteine, which is then involved, with glycine and glutamate, in GSH biosynthesis, a key non-enzymatic player in cellular antioxidant defense. Cysteine availability is the rate-limiting amino acid for GSH synthesis; therefore, cysteine uptake mediated by system Xc^−^ is important to maintain cell redox homeostasis [41,42]. Specifically, the importance of GSH is due to its thiol group; the compound is a cofactor of glutathione peroxidase (GPx) [3]. In mammals, eight GPx1-8 are known. In most cases, GPx are selenoproteins, except for GPx 5,7,8 and GPx6 in rodents [43,44]. GPx catalyzes the reduction in hydroperoxides (e.g., H_2_O_2_) in H_2_O, but also lipid peroxides (LOOH) into their respective alcohols via oxidation of reduced GSH into its disulfide form (GSSG). Specifically, GPx4 is recognized as a key mediator of a variety of human diseases, including the regulation of ferroptosis [45]. The depletion of GSH or GPx4 causes an increase in lipid peroxides that damage the cell membrane and lead to ferroptotic cell death. In fact, the presence of GSH allows a rapid intracellular degradation of hydroxides that inhibits the generation of lipid ROS; in addition, it has been shown that the use of a GPx4 activator can reduce ROS and inhibit ferroptosis [46,47]. It should be outlined that among the three different GPx4 isoforms, a cytosolic, a mitochondrial, and a nuclear one, only the cytosolic possesses these abilities. Probably, ferroptosis inhibition by GPx4 is linked to the cytosolic dislocation of the protein, which allows it to catalyze the reduction in complex hydroxides such as phospholipid and cholesterol hydroperoxides, protecting the membrane from the chain of lipid peroxidation reactions that could lead to ferroptosis [48]. In addition to System Xc^−^, cells use a minor player to produce cysteine: the transsulfuration pathway. Cysteine through this pathway can be synthesized using methionine as a precursor from which to take sulfur atoms and transfer them to serine [49]. This process may represent a compensating pathway to avoid the loss of cysteine import when the system Xc^−^-GPx4 pathway is inhibited, and in the same way, it may be considered an alternative antioxidant defense. Other than these antioxidant systems, ferroptosis may be regulated by independent pathways such as FSP1/CoQ_10_, DHODH/CoQ_10_, and GCH1/BH4. In particular, the flavoprotein apoptosis-inducing factor mitochondria-associated 2 (AIFM2), renamed as ferroptosis suppressor protein 1 (FSP1), is a CoQ_10_ redox enzyme dependent on nicotinamide adenine dinucleotide phosphate (NADP) [50]. FSP1 acts on ferroptosis through a mechanism mediated by ubiquinone or coenzyme Q_10_ (CoQ_10_). The reduced form of CoQ_10_, ubiquinol (CoQH_2_), is a lipophilic free radical trapping agent which traps peroxyl radicals and limiting lipid peroxidation, whereas FSP1 catalyzes the regeneration of CoQ_10_ using NAD(P)H [51]. In this way, FSP1/CoQ_10_ may be considered as a stand-alone parallel pathway, which co-operates with GPx4 and GSH to suppress ferroptosis [52]. A further support to this system is given by dihydroorotate dehydrogenase (DHODH), a flavin-dependent enzyme which, in the mitochondrial membrane, catalyzes the oxidation of dihydroorotate (DHO) to orotate (OA) in the pyrimidine synthesis pathway (essential for RNA/DNA synthesis) [53]. In this reaction, CoQ_10_ acting as an electron acceptor from DHODH and complexes I and II is reduced to ubiquinol (CoQH_2_) and transports electrons to complex III. Therefore, there is a synergistical action between DHODH and FSP1 involving the conversion of CoQ_10_ to CoQH_2_ which affects ROS content and improves mitochondrial functionality [54,55]. The action of DHODH restores peroxide-damaged mitochondrial lipids and inhibits ferroptosis machinery [56]. Recent studies describe the GCH1/BH4 pathway as an endogenous antioxidant enzyme system that acting independently from GPx4/GSH inhibits ferroptosis [57]. Tetrahydrobiopterin (BH4) is a cofactor involved in the redox reaction, and in the metabolism of aromatic amino acids, neurotransmitters, and nitric oxide, it plays many physiological roles including the regulation of oxidative stress and inflammation [58,59]. Its synthesis is regulated by GTP cyclohydrolase I (GCH1) [60]. BH4 is also a potent radical-trapping antioxidant, and its action is important in cases of GPx4 inhibition to protect lipid membranes from autoxidation. Hu et al. (2022) demonstrated that GCH1/BH4 inhibition causes a dangerous increase in free irons and ferritinophagy activation [61], while GCH1 overexpression reduces lipid peroxidation and protects against the ferroptosis process [62,63].

#### Erythroid Nuclear Factor 2-Related Factor and p53

Nuclear factor erythroid 2-related factor (Nrf2) is an antioxidant agent whose action may contribute to the onset of ferroptosis. In normal conditions, Nrf2 is bound to Kelchlike ECH-associated protein 1 (Keap1) in the cytoplasm; when the cell oxidative status is altered and oxidative stress increases, Nrf2 detaches by Keap1 and translocases in the nucleus. In the nucleus, Nrf2 interacts with the antioxidant response elements (ARE) mediating transcriptional activation of its responsive genes and modulating in vivo defense mechanisms against oxidative damage [64]. In detail, Nrf2 activity counteracts the increased ROS production in the mitochondria and influences mitochondrial biogenesis by maintaining the levels of nuclear respiratory factor 1 and peroxisome proliferator-activated receptor γ coactivator 1α, as well as by promoting purine nucleotide biosynthesis [65,66]. Furthermore, Nrf2 promotes GSH biosynthesis and NADPH regeneration, which are both essential to optimize GPx4 function, and regulates iron and lipid metabolism, both implicated in the ferroptosis process [67,68]. On the other hand, the p53 protein is a transcription factor mainly known as a tumor suppressor. Under physiological conditions, p53 protein levels are maintained by rapid degradation through ubiquitin-dependent proteolysis, but in response to stress signals, p53 becomes resistant to MDM2-mediated degradation and rapidly accumulates in the cell [69]. Inhibition of p53 degradation and its stabilization induces several transcriptional programs in stressed cells, including cell cycle arrest and apoptosis [70,71]. In detail, MDM2 proteins act as ubiquitin ligase shuttles which transport p53 from the nucleus to the cytoplasm, where p53 degradation takes place [72,73]. Recently, intriguing features correlate p53 activation with ferroptosis regulation, as activation of p53 significantly reduces the expression of SLC7A11 in cells [74,75]. In the cell, inhibition of SLC7A11 expression by p53 causes a decreased activity of the system Xc^−^ which, in turn, leads to decreased GSH biosynthesis and indirectly GPx4 inhibition. All these features lead the cell to an oxidative imbalance with increased lipid peroxidation culminating in ferroptosis induction [76]. However, activation of p53 may also modulate the ferroptosis process by enhancing the expression of diamine acetyltransferase 1 (SAT1) and the mitochondrial glutaminase GLS2, which are important regulators in polyamine and glutamine metabolism, respectively [77,78]. Furthermore, p53 may suppress ferroptosis, targeting the inhibition of dipeptidyl peptidase 4 (DPP4) activity or by the induction of cyclin-dependent kinase inhibitor 1A (CDKN1A/p21) expression [79]. Although these results highlight the multifunctionality of p53 that makes this protein a crucial regulatory point, a better understanding of the mechanisms by which p53 controls ferroptosis is needed.

## 4. Ferroptosis Implication in Diseases

As already mentioned, ferroptosis may have different physiological and biochemical functions within our body; it plays a beneficial role in maintaining normal physiological functions of the body as it allows the removal of damaged cells. However, ferroptosis can also result in the death of undamaged cells, triggering the progressive development of different diseases, including cardiovascular diseases, neurodegenerative diseases, and cancer (see Figure 2) [80].

### 4.1. Cancer

Cancer is a multifactorial disease that is widespread throughout the world despite significant efforts and notable improvements in the healthcare field. Overall, there are two significant challenges in treating the disease. The first concerns how to effectively kill tumor cells without harming healthy cells and the second is related to the ability of tumor cells, in the advanced stage of the disease, to resist pharmaceutical treatments [81]. One of the most popular strategies in the treatment of the disease is related to the activation of the apoptotic pathway, which results in the death of tumor and non-tumor cells. However, activation of the pathway is not always possible because cancer cells are mutated cells, and they can be resistant to certain drugs that trigger this pathway [82]. In this context, it is important to emphasize the existence of several forms of cell death because it is possible that drug-resistant cancer cells may be more susceptible to other types of cell death, including ferroptosis. The latter proves to be efficient against various types of chemo-resistant tumor cells that have escaped pharmacological treatment or are in an advanced pathological state [83]. Induced ferroptosis, in chemotherapy treatment, can be intended either as monotherapy or to improve the activity of other tumor drugs. Erastin (classic inducer of ferroptosis), for example, can increase the activity of classic tumor drugs such as temozolomide and cisplatin [84,85]. Therefore, a reduction in targets of Nrf2 (an upstream transcriptional regulator of SLC7A11 and GPx4) makes cells susceptible to the activity of proferroptotic agents in some types of cancer [86]. Different studies have shown that proferroptotic compounds target key elements of iron metabolism, such as the Xc^−^ and GPx4 systems, causing iron dyshomeostasis and triggering ferroptosis, leading cancer cells to their death. In particular, the reduced expression of genes, which encode for Xc system proteins, such as SLC7A11 and SLC3A2 causes the increase in ROS-mediated lipid peroxidation and subsequent ferroptosis, even in tumor cells that are more resistant to drug treatments [17,87]. In addition, another critical mediator of the proferroptotic cascade is acyl coenzyme A synthase long-chain member 4 (ACSL4); this enzyme, belonging to the ACL family, preferentially catalyzes the conversion of PUFAs, such as arachidonic acid, into fatty acyl-CoA esters. The catalytic activity of ACSL4 enriches the phospholipid bilayer of the cellular membrane with long unsaturated fatty acid, the main target of lipid peroxidation by ferroptosis [88]. Some studies highlighted a correlation between ACSL4 expression and cancer aggressiveness and proliferation [89,90,91]; subjects with colorectal cancer with a high expression of ACSL4 have a poor prognosis and lower survival rate, while subjects with brain, breast, or lung cancer with lower ACSL4 expression have a more hopeful prognosis [91]. Several studies also tested the connection of ferroptosis with immunotherapy and radiotherapy in cellular cancer death. In detail, immunotherapy results in the activation of the antitumor immune response activated by cytotoxic T cells. Interferon gamma (IFN-γ) released by CD8 T cells can hinder the uptake of cystine by tumor cells as it results in under regulation of the expression of SLC3A2 and SLC7A11 (subunits of the system Xc^−^) and thus induces lipid peroxidation and ferroptosis in ovarian carcinoma, melanoma, and fibrosarcoma [5,92]. On the other hand, radiotherapy is a common modality of cancer treatment in which targeted delivery of ionizing radiation is used to eradicate cancer cells. Ionizing radiation induces DNA damage and stimulates ROS generation, provoking a high lipid peroxidation state, which is one of the main ferroptosis features.

Indeed, biochemical evidence correlates ferroptosis with cancer radiotherapy, which results in ACSL4 expression triggering PUFA peroxidation and ferroptosis in different tumors, including melanoma and esophageal cancer [93,94]. Collectively, several studies support the key role of ferroptosis and lipid peroxidation in preclinical and clinical settings to enhance and potentiate the effectiveness of chemotherapy drugs. However, further studies are needed to better elucidate this synergistic mechanism and to explore the many key regulations of ferroptosis.

### 4.2. Neurological Diseases

Neurological disorders are the leading cause of disability in the world; and they include a wide variety of pathological conditions; the most known are Alzheimer’s (AD), Parkinson’s (PD), Huntington’s disease (HD), and amyotrophic lateral sclerosis (ALS). A common feature of neurodegenerations is an accumulation of altered proteins and progressive loss of neurons. This condition is generally related to an inflammatory local state due to iron accumulation, oxidative stress, and lipid peroxidation. Furthermore, high oxygen consumption makes the brain, rich in PUFAs, particularly vulnerable to triggering the ferroptosis process [95,96].

#### 4.2.1. Alzheimer’s Disease

Alzheimer’s disease (AD) is a widespread neurodegenerative disease generally characterized by extracellular amyloid protein deposition and intracellular neurofibrillary tangles due to hyperphosphorylation of the tau protein. These features cause a reduction in neurogenesis, neuronal death, and loss of synaptic connections. Scientific evidence has shown iron accumulation in the brain of patients with AD, positively correlated with disease progression and cognitive decline [97,98]. In AD brains, the imbalance of iron homeostasis is closely related to amyloid plaques and neurofibrillary tangles [99]. Iron deposition seems to colocalize in areas of the brain where neurofibrillary tangles are found, and this condition is associated with the progression of neurodegeneration [100]. Increased lipid peroxidation and abnormal iron accumulation in AD brains are the two trigger conditions of ferroptosis [101]. In AD mice with a GPx4 deficiency, it has been demonstrated that ferroptosis inhibitors improve the condition of neurodegeneration by blocking excessive iron accumulation and lipid peroxidation, thus reducing oxidative stress and inflammation [102,103]. Recent in vivo measurements of the brain and blood GSH in AD have revealed decreased GSH levels [104]. Furthermore, AD is characterized by iron dyshomeostasis, an altered expression of the Xc^−^ system, and by increased lipid oxidation, suggesting that therapies targeting ferroptosis may be potentially beneficial [105]. This suggests that regulation of iron metabolism and reduction in neuronal ferroptosis, with ferroptosis inhibitors, may be a promising therapeutic approach. Among the molecules involved in this mechanism, deferoxamine (DFO) is a synthetic iron chelator and is commonly used in the clinical setting; however, it is important to note that it shows 50% efficacy and fails to cross the blood–brain barrier. Another compound used in the treatment of the condition is deferiprone (DFP). This, like the previously mentioned molecule, is a synthetic compound that acts as an iron chelator. Compared with DFO, DFP shows greater efficacy and a safer therapeutic profile; in addition, this compound appears to be able to cross the blood–brain barrier. Quinoline and its derivates are also iron chelators [106]. These molecules, from studies conducted in animal models by Grossi et al., and Crounch et al., appear to induce an improvement in cognitive function and result in a reduction in the accumulation of Aβ by determining its degradation [107,108]. In addition, studies conducted by Wang et al. demonstrated that clioquinol results in a reduction in the expression of secretase β and γ and amyloid precursor protein (APP), while Lin et al. proved the ability to reduce tau protein tangles via tau degradation [109,110]. In addition to the previously mentioned molecules, another compound that can inhibit the ferroptotic process in AD is hepcidin. This can reduce the transport of iron across the blood–brain barrier, thus inhibiting its accumulation. In a study conducted by Du et al., 2015, it was seen that this compound in microvascular endothelial cells significantly inhibits the expression of FPN1, TfR1, and DMT1, can regulate iron homeostasis at the neuronal level, reducing iron uptake and release, and can also reduce Aβ plaque formation by increasing cognitive activity in the mouse models used in the study [111,112].

#### 4.2.2. Parkinson’s Disease

Parkinson’s disease (PD) is manifested by muscle rigidity, tremors, even during rest, difficulty in starting or finishing movements, and balance disorders. The two main pathological features of PD are progressive dopaminergic neurodegeneration in the substantia nigra and misfolded a-synuclein, the major protein component of Lewy bodies. Scientific studies have highlighted the connection between the progression of PD and iron homeostasis in the brain tissue, showing common features with ferroptosis related to increased ROS generation, GSH depletion, and lipid peroxidation, which lead to the accumulation of malondialdehyde, a toxic subproduct [113,114]. Iron ion accumulation is correlated with an increased risk of the formation of α-synuclein fibers because of iron ions’ high affinity for α-synuclein. Iron accelerates the deposition of the misfolded protein, contributing to PD degeneration [115]. Furthermore, increased lipid peroxidation due to iron accumulation in the cell leads to damage of dopaminergic neurons, decreased dopamine production, and disruption of the neuron’s integrity [116]. In t-BHP-induced PC12 cells, used as a model of PD, a decrease in GPx4 expression, reduction in the GSH/GSSG ratio, and increase in lipid peroxidation were found; the harmful condition was reversed by ferrostatin-1 and deferoxamine, indicating an iron dependence of the process [117,118]. In this context, chelating agents are very important in pharmacological treatment for neuroprotection because iron chelation preventing Fe entrance in the Haber–Weiss reaction reduces ROS formation and oxidative stress [116,119,120]. An important chelating agent is DFP, which shows neuroprotective activity in patients with early-stage Parkinson’s disease [119]. In addition to this molecule, it was observed by Billings et al., in a study in transgenic PD mouse models, that clioquinol has the ability, as an iron chelator, to reduce the loss of neurons from the substance nigra, thereby protecting neurons from iron accumulation and inhibiting ferroptotic processes [120].

#### 4.2.3. Huntington’s Disease

Huntington’s disease (HD) is a very rare disease that affects the central nervous system and is characterized by short and sudden involuntary movements, psychiatric disorders, dementia, and behavioral disorders. It is an autosomal dominant disorder caused by an abnormal repeat of a sequence of three DNA bases, cytosine–adenine–guanine (CAG) expansion in the huntingtin gene on chromosome 4. The mutated gene encodes an expanded polyglutamine stretch in the Huntington protein (Htt); longer CAG repeats are the cause of earlier onset and more severe symptoms in patients [121,122]. Mutant Htt protein accumulation is the trigger for pathological progression of the disease because of its aggregation tendency to form macromolecules, which leads to neuronal death. Cytoplasmic and nuclear toxic fragments cause mitochondrial dysfunction and increase ROS generation [123]. This status is worsened by lower GSH and GSH-S transferase levels detected in HD brains that cause an imbalance between antioxidant defenses and pro-oxidant conditions, increasing oxidative stress [124]. As a result, high lipid peroxidation is one of the main features in HD patients and increased formation of lipid peroxides colocalized with mutant Htt protein inclusions in HD mouse models has been detected [125,126]. In a recent article, Song et al. indicated arachidonate 5-lipoxygenase (ALOX5) as a mediator for the ACSL4 ferroptosis process in HD [127]. ALOX5 is a member of the lipoxygenase family, which metabolize arachidonic acid as well as PUFAs and contribute to lipid peroxidation [128]. In HD mice, the loss of ALOX5, by inactivation of Alox5 gene, ameliorated their pathological phenotypes, extending the life spans [113]. Iron dysmetabolism and accumulation is another characteristic in HD patients and a pivotal trigger of ferroptosis [129,130,131,132]. The implication of ferroptosis in HD is also supported by the positive action of iron chelators. Chen et al. demonstrated in R6/2 HD mice that intraventricular administration of deferoxamine improved the striatum pathology and motor phenotype [131]. Inhibitors of ferroptosis, such as ferrostatin-1 and liproxstatin-1, are also very important in the treatment of HD; they act by inhibiting damage caused by lipid peroxidation, thereby preserving healthy neurons [80]. The activity of Fer-1 was demonstrated through a study conducted by Skouta et al., where it was seen that treatment with Fer-1, at different concentrations of 10 nM, 100 nM and 1 μM, protects neurons from degeneration; in the same study conducted in MH cell models, it was seen how the molecule counteracts lipid peroxidation [133]. Another molecule that would seem to have effects in HD is DFO. A study conducted by Yang et al., in murine models of MH, demonstrated the protective activity of this compound, also improving the cognitive ability of the study models [134].

#### 4.2.4. Amyotrophic Lateral Sclerosis

Amyotrophic lateral sclerosis (ALS) is a neurodegenerative disorder caused by selective degeneration of motor neurons in the brain and spinal cord. The disease leads to a slow but progressive decrease in the ability to perform voluntary movements and to progressive paralysis [135]. Despite fervent studies on ALS, to date the mechanism related to the onset of the disease remains partly obscure. Several genes have been identified as possible causes of ALS onset, including mutations in the gene for Cu, Zn superoxide dismutase 1 (SOD1) responsible for 20–40% of ALS familial forms [136]. Recently, a correlation has been discovered between ferroptosis and motor neuron death in ALS. Decreased levels of GPx4 have been found in the spinal cord and brain of ALS mouse models and in the post-mortem spinal cord of ALS patients [137]. Magnetic resonance imaging also found iron accumulation in the motor cortices of some ALS patients and in spinal cords of ALS animal models [138]. Low levels of GSH and high lipid peroxidation levels detected in the red blood cells of ALS patients confirm an alteration in oxidative state that is very similar to the ferroptosis mechanism [139]. From different studies found in the literature, several molecules have emerged that can intervene in the modulation of ferroptosis to reduce the progression of the disease. In a study conducted by Chen L. et al., on NSC-34 cells expressing the SOD mutant, Fer-1 was seen to be able to increase cell viability by inhibiting lactoperoxidase (LPO) activity and resulting in a reduction in lipid ROS; furthermore, in addition to Fer-1, it has been also observed that lipro-1 is able to inhibit LPO activity, resulting in increased cell viability [140]. On the contrary, studies conducted by Devos D. et al. on transgenic mice have found that deferiprone is a molecule able, through inhibition of LPO, to reduce iron accumulation in the spinal cord, increase the survival of study models, and reduce the markers used for oxidative phenomena [141]. Vitamin E also appears to be involved in inhibition of the ferroptotic pathway, promoting the attenuation of ALS. In fact, in a study by Chen L. et al. it was seen that the vitamin, in GPX4NIKO mouse models, not only delays the onset of paralysis but also manages to delay the death that is induced by GPX4 excision [142].

### 4.3. Cardiovascular Diseases

Scientific evidence has shown that altered iron homeostasis and ferroptosis are closely related to cardiovascular diseases as the accumulation of iron in the myocardium causes cardiotoxicity and triggers the alteration of normal cardiac function [143,144]. Additionally, Fang et al. showed that in the heart, doxorubicin (also known as Adriamycin, an anthracycline antibiotic with antineoplastic action that has a broad spectrum of antitumor activity) can trigger ferroptosis and free iron released from heme degradation can cause cardiac damage [145]. However, it is important to emphasize that even an iron deficiency can be harmful to cardiomyocytes as it can lead to an alteration in the contractile and metabolic activity of the cells themselves [146]. It is therefore stressed in hearts the importance of a regular and modulated iron homeostasis in which hepcidin is the main regulator. This protein, synthesized in the liver, can inhibit the activity of ferroportin, the only known iron exporter. Regulation of ferroportin is critical for iron homeostasis; the inhibition by hepcidin results in an iron decrease, while hepcidin depletion, which can be due to genetic mutations or blood transfusions, causing iron overload can trigger ferroptotic processes [147]. This state can be therapeutically partially restored by iron chelators or genetic regulation of ferroptosis by acting on specific targets such as SLC7A11 [148,149]. In this context, a promising strategy to counteract the accumulation of free iron may be represented by iron chelators, such as deferoxamine and deferiprone, which counteract the overload of circulating metal [145,150].

### 4.4. Ischemic Reperfusion Injury

Ischemia is an imbalance between the supply (perfusion) and the demand for oxygenated blood to a tissue or organ which leads to consequent functional, biochemical, and morphological alterations, followed by potential cellular necrosis of the affected area. The restoration of blood flow in an ischemic tissue can lead to an exacerbation of the damage, a condition known as “ischemic reperfusion injury” (IRI). IRI contributes to mortality and morbidity in many pathological conditions, including myocardial infarction and stroke, and occurs with a variety of surgical interventions where blood flow is temporarily blocked but then reperfused. The IRI process is multifactorial and not yet fully elucidated. In general, its pathogenesis involving different mechanisms can be divided into two different steps: the ischemic state and the reperfusion state. During ischemia, the interruption of the oxygen supply causes the blockage of mitochondrial oxidative phosphorylation and the consequent drastic reduction in ATP production, which becomes almost complete (about 90–95%) after about 40–60 min [151,152]. There is also an increase in anaerobic glycolysis that generates lactate and causes a change in intracellular pH that drops to a value of 6.2 after about ten minutes [153]. An ionic imbalance is therefore established also due to the blockage of the Na^+^/K^+^ and Na^+^/Ca^2+^ pumps; Na^+^ and Ca^2+^ accumulate inside the cell which, combined with the accumulation of other metabolites such as lactate and protons, draws water inside the cell, causing swelling [154,155]. Reperfusion provides oxygen and allows the replenishment of substrates in the ischemic area that are essential to produce ATP, glucose, and fatty acids, all of which are important for cell survival but can also contribute to exacerbating ischemic damage. In fact, during reperfusion, the enhancement of mitochondrial respiration leads to the generation of large amounts of ROS which, accompanied by upregulation of oxidative enzymes including xanthine oxidase, nicotinamide adenine dinucleotide phosphate oxidase (NOX), cyclooxygenase, lipoxygenase, and iNOS, cause a complex inflammatory response, leading to exacerbation of ischemic injury [156,157]. Recently, ferroptosis has been studied as a potential contributor to the events triggering ischemic injury [11]. In detail, during the reperfusion phase and not in the ischemic one, several key events of ferroptosis occur, including ROS exacerbation, increased intracellular iron and malondialdehyde concentrations, lipid peroxidation, reduced mitochondrial volume, and reduced or lost mitochondrial cristae [156]. Therefore, the synthesis or discovery of new molecules capable of antagonizing oxidative stress and the ferroptosis molecular targets may contribute to alleviating IRI and suppressing the inflammatory response. Chen et al., 2023, demonstrated that synthesized chitosan-derived nitrogen-doped carbon dots (CNDs), a synthetic product with ROS scavenging capabilities, protected the liver against IRI by suppressing oxidative stress damage [158]. As mentioned earlier, IRI can affect different organs of the body; in fact, we observe that in addition to the liver, the pathology can also affect the brain, kidneys, lungs, myocardium, etc., and obviously the inhibitory action of the different molecules towards the pathology and ferroptosis will be different. Wu J.R. et al. in their study showed how ischemic stroke results in the degeneration of the tau protein (protein involved in neuronal microtubule formation and iron export), causing iron accumulation and worsening of IRI symptoms. Scholars have also seen that there are molecules that can intervene in this condition, inhibiting the progression of the process; examples are liprostatin-1 and ferrostatin-1 [159]. At the cardiac level, on the other hand, myocardial reperfusion injury results in the death of cardiac myocytes, and it has also been seen in clinical studies that myocardial iron is a risk factor as it results in left ventricular remodeling after reperfusion [160]. Fang et al. in their study demonstrated how the intervention of ferrostatin-1, a synthetic inhibitor of ferroptosis that can both reduce lipid peroxidation and act as a scavenger, is of paramount importance in treating the condition; iron chelation can also result in improvements in acute and chronic IRI [145]. IRI can also affect the kidneys, in which case what is observed is an increase in reactive oxygen species, a direct involvement of iron, which as it accumulates triggers the ferroptotic process, resulting in renal tubular cell death [161]. However, it is important to emphasize that in renal IRI, one cannot exclusively target ferroptosis but one must also pay attention to necrosis. In a study conducted by Pefanis et al., it was seen that there is a direct correlation between the two pathways and therefore inhibition of only one pathway is not sufficient to result in improvement [162]. Thus, it follows from these studies how IRI might be related to ferroptosis and how the use of chelating agents and ferroptosis inhibitors may counteract the progression of the disease.

## 5. Ferroptosis Modulators

The association of ferroptosis with many diseases and the possibility of counteracting tumor growth and reducing drug resistance have contributed to a significant growth of interest in ferroptosis as potential therapeutic target (see Table 2 and Table 3). In this context, the continuous discovery of new drugs able to induce or inhibit ferroptosis is crucial to the research. There are many ways to induce ferroptosis, but the main systems involved in the process are iron homeostasis, the system Xc^−^, and GPx4 catalytic action, and in some cases the action directed on one of these systems may trigger the blockage or activation of another also implicated in the process. Among the molecules that may modulate the death process, RAS-selective lethal 3 (RSL-3) is a compound that selectively inhibits the activity of GPx4, leading to downregulation of some factors involved in ferroptosis, such as Activating Transcription Factor 4 (ATF4) and SLC7A11 [87,163]. GPx4 inhibition leads to the imbalance of oxidative defenses and an increase in oxidative stress that consequently leads to ferroptosis; thus, inhibitors of GPx4 pathway can be exploited against highly aggressive forms of cancer such as glioblastoma, one of the most malignant brain tumors [164]. In addition, Erastin can also act directly on the system Xc^−^ through selective inhibition of this pathway, which will result in reduced cystine cellular uptake. The consequent reduced availability of cysteine will lead to the depletion of GSH, indirect inhibition of GPx4, and an increase in lipid peroxidation [2]. Overall, erastin has demonstrated multipotential properties that make it a selective candidate for cancer treatment as a chemotherapeutic drug to enhance the sensitivity of cancer cells [165,166]. Another synthetic inducer of ferroptosis is FIN56, which also acts by causing the permeabilization of the lysosomal membrane (LMP) [167]. FIN56 derives from a class of molecules called “caspase-3/7-independent lethals” (CILs) that have been shown to induce cell death through ferroptosis and necrotic death [168]. Within this category, CIL56, named FIN56, is the compound that has shown the greatest activity in inducing ferroptosis at low concentrations. FIN56 acts by promoting GPx4 protein degradation and thereby causing ferroptosis [169]. Some experiments on tumor cell lines have shown that after treatment with FIN56, although the amount of GPx4 decreases, its transcriptional levels increase, demonstrating that FIN56 does not act at the transcriptional level but in the post-translational phase, causing a depletion of GPx4 [170]. Furthermore, FIN56 activates squalene synthase, an enzyme directly involved in the production of cholesterol, and this leads to the blockade of some natural antioxidant systems such as coenzyme Q10 (CoQ10) and Selenocysteine-tRNA (Sec-tRNA) [87,169]. Besides FIN56, other inhibitors of GPX4 are Bufotaline (BT) and palladium–pyrithione complex (PdPT). In a study conducted by Zhang W. et al., it was seen that BT can trigger the ferroptotic process by inhibiting GPX4. The investigators used the A549 cell line (lung cancer cells) and saw that the compound inhibited cell proliferation by inducing ferroptosis. Through different techniques, such as immunoblotting and the immunoprecipitation assay, it was seen that bufotaline inhibited not only the protein expression of GPX4 but also induced its degradation; in addition, this molecule was also associated with the ability to increase intracellular Fe^2+^ concentration and induce lipid peroxidation [171]. PdPT activity, on the other hand, was observed by Li Y. et al. in a study conducted on NSCLC A549 and NCI-H1299, two cancer cell lines. The investigators saw that the palladium–pyrithione complex has the ability to inhibit deubiquitinases (DUBs), enzymes critical for cell metabolism and survival; furthermore, inhibition of DUB resulted in the activation of caspases, triggering the apoptotic pathway, and degradation of GPX4, triggering ferroptosis, suggesting the involvement of the complex in the ferroptotic process [172]. Ferroptosis regulators can be divided based on the molecular mechanism by which they act. In this context, Deferoxamine is a chelating agent used for iron and aluminum toxicity that binds to free iron in the body, making it available for elimination through urine and feces. By reducing the availability of free iron within cells, deferoxamine prevents ROS production and lipid peroxidation [2]. Deferoxamine has been demonstrated to inhibit ferroptosis, promoting recovery of traumatic spinal cord injury and protecting liver cells from induced cell death in a model of acute liver necrosis [6,173]. Another well-known iron chelator is Deferiprone (DFP, Ferriprox), orally administered and approved by the FDA for the treatment of patients with iron overload. DFP, as deferoxamine, reducing free iron levels may be considered a promising therapeutic drug which prevents oxidative stress, resulting in a decreased inflammation state [174,175]. Between the ferroptosis inhibitors, Ferrostatin-1 is a synthetic antioxidant able to generate a complex with iron, preventing oxidative stress from the Fenton reaction [96]. Specifically, ferrostatin-1 may act as a scavenger; binding with hydroperoxyl radicals, it breaks the peroxyl chain mechanism produced by ferrous iron [133,176]. Similarly, to ferrostatin-1, Liproxstatin-1 (Lipro-1) can block ferroptosis, even in the presence of known ferroptosis inducers such as FIN56 or RSL-3 [75,177]. Fan et al. in a study on an oligodendrocyte (OLN-93 cell line) model of ferroptosis induced by RSL-3 demonstrated that concentrations between 115.3 nM and 1 μM of Lipro-1 reduced lipid peroxidation, improved GPx4 expression, and increased GSH levels [178]. The ferroptosis modulators described above are semisynthetic derivatives (for a broader overview refer to Table 2 and Table 3), but growing evidence highlights natural products as a potent drug-discovery resource and some of them such as polyphenols can induce or inhibit programmed cellular death [179,180,181].

## 6. Natural Ferroptosis Regulators

Natural compounds are not only safer, as they are less toxic, but they act more specifically on the targets involved in ferroptosis and are also more accessible and economical [182]. These compounds can act directly on target cells by inducing oxidative stress or can affect cellular transport and metabolism of iron and its homeostasis or cause GSH depletion, leading to the formation of free radicals and damage to cell membranes, which are typical events of ferroptosis (see Table 2 and Table 3).

### 6.1. Natural Inducers

Among natural inducers, Artemisinin, a naturally occurring compound extracted from *Artemisia annua*, is a sesquiterpene consisting of three isoprene units linked together and presenting a peroxide bridge within the structure [183]. Typically, artemisinin and its derivatives are used as antimalarial drugs and are being studied for their potential use in the treatment of other diseases, including cancer [184]. In addition to the strong antimalarial activity, artemisinin, thanks to its chemical structure and the peroxide bridge, may affect iron metabolism and the Xc^−^/GPx4 axis, causing an increase in ROS generation, which is the basis of ferroptosis [185]. Among the artemisinin derivatives, we can also consider artesunate and artemether, both ferroptosis inducers. In the first case, artesunate promotes the hydrolysis of ferritin, increasing the amount of free iron ions and contributing to the imbalance of intracellular iron metabolism. The intracellular increase in free iron through the Fenton reaction improves ROS generation, lipid peroxidation, and promotes ferroptosis in pancreatic ductal adenocarcinoma. The second derivate, artemether, acts on several ferroptosis targets, including increased iron levels and lipid peroxidation products, decreased GSH, and the p53 pathway. Specifically, p53 is crucial for artemether-induced ferroptosis because artemether not only promotes the expression but also the nuclear import of p53, which, in turn, reduces the expression or causes a decreased activity of the system Xc^−^ in cells [186]. Among other natural inducers, we can also consider isothiocyanates (ITCs), a class of sulfur-containing organic compounds derived from thiocyanic acid and containing the functional group -N=C=S. These compounds, present in some plants of the Brassicaceae family, such as broccoli, cabbage, cauliflower, and radishes, are known to have antitumor, antimicrobial, and anti-inflammatory properties [187]. Among all, the one that exhibits the highest antitumor activity is phenethyl isothiocyanate (PEITC). In fact, it has been shown that this compound is able to block the growth and survival of some tumor cell lines. Moreover, through a mechanism that is not yet fully defined, it induces ROS production and ferroptosis in several cellular lines [188]. Kasukabe T. et al. demonstrated that in pancreatic cancer, the combined treatment of PEITC with Cotylenin A (CN-A), a natural antitumor drug, has a synergistic antiproliferative activity by inhibition of the growth of MIAPaCa-2, PANC-1, and gemcitabine-resistant PANC-1 cells [189]. Another compound with already known apoptotic and antitumor activity is Gambogic acid (GA). This compound, mainly extracted from G. hanburyi and G. morella, well-known plants in Chinese culture, not only blocks cell proliferation but scientific evidence shows it also inhibits invasion, metastasing processes, and angiogenesis in various tumor cell lines [190,191,192,193]. GA induces apoptosis and autophagy through multiple mechanisms and perturbs the cellular redox balance, inducing ROS generation and ferroptosis death in tumor cells [194].

### 6.2. Natural Inhibitors

Through screening of natural compound libraries, it has emerged that Glycyrrhizin (GLY) is able to block ferroptosis. GLY is an organic compound found in the root of liquorice (Glycyrrhiza glabra) that gives the plant its sweet taste. It is a mixture of triterpenoid saponins and has been the subject of numerous studies for its pharmacological properties, including anti-inflammatory, antioxidant, immunomodulatory, and antiviral properties [187,195]. Wang et al. have shown that GLY reduces ferroptosis in hepatocytes during acute liver failure (ALF) by activating various pathways, such as Nrf2/HO-1/HMGB1, to counteract oxidative stress involved in ferroptosis [196]. There are some ferroptosis modulators that do not act directly on iron metabolism or on the main mechanisms underlying this type of cell death but act indirectly on other pathways that, in parallel, can affect the ferroptosis process. In this context, several pathological conditions involving the inflammatory process may be worsened by the activation of the JAK-STAT pathway, which indirectly leads to ferroptosis. Indeed, the binding of cytokines such as TNF and IL-6 to their receptors causes the dimerization and phosphorylation of JAK, which in turn activates the STATs; particularly, STAT3 acts at the nuclear level increasing the expression of hepcidin, which inhibits iron export, while STAT1 activation inhibits the system Xc^−^, with both conditions resulting in ferroptosis [197]. Gao et al. studied the indirect correlation between the JAK-STAT pathway, ferroptosis, and tumor progression in certain cell lines. Ferroptosis was induced by incubating PDAC cell lines with erastin, and then Cryptotanshinone, an inhibitor of the process, was added. Cryptotanshinone is a natural chemical compound belonging to the tanshinone class, isolated from the root of Salvia miltiorrhiza; the study demonstrated that Cryptotanshinone can block erastin-induced ferroptosis by antagonizing STAT3 [198]. In addition, N-acetylcysteine (NAC), a chemical agent with a thiol group derived from L-cysteine, acting as an antioxidant agent alleviates ferroptosis and reduces cell death [199]. NAC not only blocks the production of some inflammatory cytokines such as TNF-α and IL-1β, also blocking the JAK-STAT pathway, but it inhibits the production of leukotrienes, inflammatory mediators implicated in various pathologies including asthma, allergies, and intracerebral hemorrhage. NAC prevents cell death by targeting nuclear ALOX5-derived reactive lipid species, which are normally precursors of ferroptosis. This action leads to a lower demand for glutathione to restore iron and lipid homeostasis, thus enabling an increase in cellular GSH levels. Moreover, NAC can act as a donor of sulfhydryl groups, further increasing GSH synthesis. An increase in GSH can help protect cells from death by ferroptosis since glutathione is an important endogenous antioxidant that can neutralize free radicals and ROS, thereby reducing oxidative stress and protecting cellular membranes from damage [200].

#### Polyphenols

A growing body of studies highlights polyphenol’s multiple effects against ferroptosis and correlates the intake of foods rich in these compounds including fruits, vegetables, and cereals to the decreased incidence of chronic and tumor diseases [201]. Generally, polyphenols due to their antioxidant and anti-inflammatory effects may act as ferroptosis inhibitors by counteracting and mitigating the underlying mechanisms of oxidative stress, although it is known that polyphenols’ activity depends on the dose, treatment duration, and cell/tissue specificity; so, it is more correct to classify them as inducers or inhibitors or both. Among polyphenols, many flavonoids have been reported to be potential ferroptosis inhibitors. In detail, flavonoids belong to the group of polyphenols and based on their chemical structure can be divided in flavones, flavonols, flavanones, anthocyanidins, isoflavones, and flavanols [202]. Curcumin, a yellow polyphenolic pigment, particularly present in the tuberized rhizome (root) of various species of turmeric, such as Curcuma longa (or Curcuma domestica), is a powerful antioxidant with beneficial properties in the treatment of oxidative stress [203]. Treatment with curcumin significantly increases nuclear transfer of Nrf2 and the expression of Gpx4 and HO-1 and inhibits glucose-induced ferroptosis in cardiomyocytes [204]. Guerrero-Hue et al. demonstrated, on mouse models with acute kidney injury, that curcumin reduces lipid peroxidation, inflammation, and renal damage associated with rhabdomyolysis by decreasing ferroptosis-mediated cell death [205]. In addition, Kose et al. observed a protective effect of curcumin treatment (20 μM for 24 h) against erastin-induced ferroptosis in mouse insulinoma pancreatic cells (MIN6). In detail, the study investigated the effects of two polyphenols, curcumin and epigallocatechin-3-gallate (EGCG), against iron loading and ferroptosis; both compounds intervene positively by acting as iron chelators and preventing GSH depletion, GPx4 inactivation, and lipid peroxidation [206]. The iron chelating activity of EGCG could be related to the molecule’s ability to modulate the expression of IRP involved in iron homeostasis. These proteins, activated under iron-deficient conditions, inhibit ferritin activity and simultaneously stabilize TfR1 mRNA. The potentiality of EGCG as a ferroptosis inhibitor is confirmed by Yang et al., who demonstrated that its administration attenuates iron metabolism disorders, increases Nrf2 and GPx4 expression, and elevates antioxidant capacity in iron overload mice [207]. An increase in antioxidant activity has been shown also by apigenin, a known flavone found in chamomile leaves and celery; it was able to block oxidative stress and confer neuroprotection against kainic acid-induced ferroptosis by inducing GPx4 and SIRT1. Shao C. et al. on human neuroblastoma SH-SY5Y cells showed that apigenin protects against ferroptosis induced by myeloperoxidase overexpression and kainic acid [208]. Another well-known flavonoid that has been correlated with ferroptosis is quercetin, a flavonol particularly found in grapes, red onions, blueberries, apples, and radicchio [209]. As other polyphenols, quercetin possesses a high antioxidant activity which gives the compound a protective action against ferroptosis. Li X. et al., after over-loading mice with a ferric–dextran complex, demonstrated that quercetin administration attenuated lipid peroxidation in hepatic, renal, and cardiac cells thanks to its iron chelating nature [210]. Furthermore, Li et al. showed that quercetin (0.03 µM), and its metabolite quercetin Diels-Alder anti-dimer, can protect bone marrow-derived mesenchymal stem cells from erastin-induced ferroptosis, possibly through the antioxidant pathway [211]. Another polyphenol involved in countering the ferroptosis process is baicalein. In a study conducted by Xie Y. et al., it was observed that baicalein exhibits protective activity in human pancreatic cancer cells (BxPc3) and epithelioid carcinoma cells (PANC1) of the pancreas. The molecule inhibits GSH depletion, GPx4 degradation, and lipid peroxidation. Furthermore, the study demonstrated that baicalein inhibits Nrf2 degradation and reduces oxidative stress in PCNA1 cells [212].

## 7. Conclusions

In recent years, an emerging enthusiasm in ferroptosis processes and the role it plays in the onset and progression of various diseases has driven numerous studies. An in-depth understanding of all the molecular mechanisms involved in this form of cell death could facilitate the identification of molecular targets more suitable for the treatment of different diseases and optimize clinical prevention, diagnostics, and clinical treatment. To date, many mechanisms of ferroptosis have not yet been fully elucidated, including the trigger of the process, which may be caused by iron accumulation, GPx4 and system Xc^−^ inhibition, ROS increase, and lipid peroxidation. It has not yet been fully clarified whether the alteration of a single factor is sufficient to trigger ferroptosis or whether the beginning of the process depends on a synergy between all these factors. In addition, ferroptosis shows some regulatory signals common to other types of cell death and this makes difficult a net distinction between the different mechanisms of death. As discussed, many synthetic and natural-origin molecules have been tested for the regulation of ferroptosis and for the treatment of possibly related pathologies, including neurodegenerative, cardiovascular, or metabolic disorders and many cancer types. It is known that the activation of ferroptosis in some types of cancer can induce the death not only of ordinary cancer cells but also of those resistant to pharmacologic treatment. This research unequivocally highlights the importance of natural compounds as potential modulators and inducers of ferroptosis, emphasizing their revolutionary prospects in the treatment of various pathologies. Delving into the interactions between these natural compounds and the regulatory mechanisms of ferroptosis paves the way for targeted therapeutic strategies, and understanding such processes can not only contribute to elucidating some different mechanisms but also guide the development of more effective and specific therapies. Indeed, plants produce a large multitude of antioxidant compounds, all potentially usable to fight the ferroptosis process and, with it, slow down or block the progression of the related disease. However, despite growing evidence pointing to natural products as ideal compounds for both single- and multi-drug treatment against ferroptosis, studies are still needed to elucidate their pharmacokinetics, bioavailability, nuclear pharmacology, and to optimize dosage for potential clinical therapy. Furthermore, this research not only enhances our understanding of the links between ferroptosis and pathologies but also opens the door to new innovative therapeutic perspectives, promoting an integrated approach to disease management through the responsible and targeted use of natural modulators of ferroptosis. Therefore, future studies should aim to clarify specific molecular targets to avoid side effects and develop drugs with highly specific molecular targets.

## Figures and Tables

**Figure 1 ijms-24-17279-f001:**
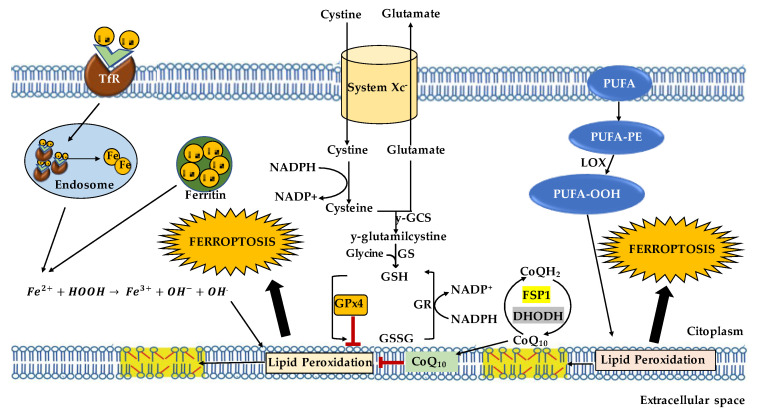
Ferroptosis pathway. Ferroptosis may be triggered by transferrin endocytosis linked to transferrin receptor 1 (TfR1). After endocytosis, transferrin releases Fe^3+^ which is reduced in Fe^2+^. Ferrous iron-mediated Fenton reaction increases ROS such as hydroxyl radical that react with membrane lipids inducing lipid peroxidation. Lipid peroxidation is also alimented via LOX action which oxidizes PUFA, generating the corresponding hydroperoxide derivatives which, in turn, react with other membrane lipids. Lipid peroxidation may be regulated by GPx4, which converts hydroperoxides in H_2_O and lipid peroxides into their respective alcohols via oxidation of GSH into its disulfide form GSSG. GSH levels are maintained by system Xc^−^, which mediates the exchange of extracellular glutamate and intracellular cystine required for its synthesis.

**Figure 2 ijms-24-17279-f002:**
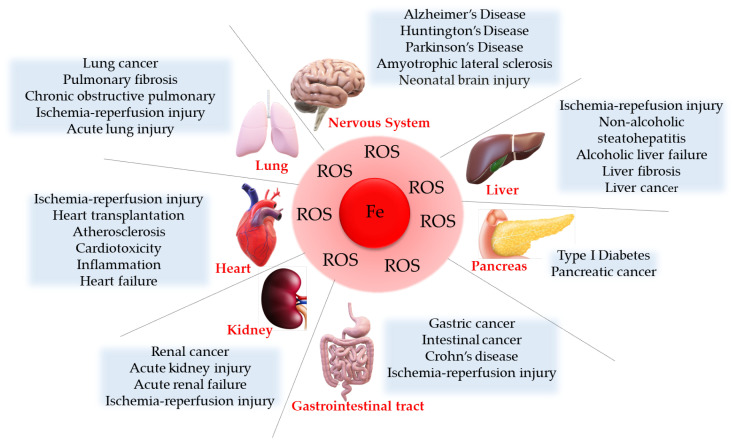
Ferroptosis-related diseases and pathophysiological implications. Ferroptosis can be associated with different diseases affecting different body districts, such as nervous system, digestive system, respiratory system, circulatory system, urinary system, and immune system.

**Table 1 ijms-24-17279-t001:** The main features of Ferroptosis, Necroptosis, Apoptosis, Autophagy, and Pyroptosis.

	Ferroptosis	Necroptosis	Apotosis	Autophagy	Pyroptosis	References
**Cell morphology**	Swelling, reduction of mithocondrial cristae	Swelling	Shrinkage, intercellular junction disappearence	Vescicles in cytoplasm, autophagosome formation	Swelling, bubbling	[1,2,4]
**Membrane**	—	Rupture plasma membrane	Membrane blebbing	—	Rupture plasma membrane	[1,4]
**Nucleus**	—	Pyknosis, Karyorrhexis, Karyolysis	Chromatine condensation and nuclear disintegration		Chromatine condensation	[1,4]
**Biochemical features**	Iron and ROS accumulation, GSH depletion, lipid peroxidation	Lower ATP level	Caspase 3, 6, 7 activation		Caspase 1, 4, 5, 11 activation	[1,2,4]

**Table 2 ijms-24-17279-t002:** Natural and synthetic inhibitors of ferroptosis.

Drugs	Nature of the Compound	Targets	Test Models	Functions	References
Deferoxamine	Synthetic	Iron	HT-1080; Calu-1; BJeLR; PC12; MEF cells; Aging model mice	Iron chelation; ROS generation inhibition; Lipid peroxidation inhibition	[2,17,152,155]
Deferiprone (DFP)	Synthetic	Iron	Patients with iron overload	Iron chelation; ROS generation inhibition	[153]
Ferrostatin-1	Synthetic	15-LOX/PEBP1	HT-1080 cells	Free radical scavenger; Lipid peroxidation inhibition	[158,159]
Lipro-1	Synthetic	—	OLN-93 cell line	Lipid peroxidation inhibition; GPx4 expression improved; GSH levels increase	[145,159,160]
Glycyrrhizin (GLY)	Natural	HMGB1/GPx4 Pathway	Hepatocytes	Lipid peroxidation inhibition	[173,174,175]
Cryptotanshinone	Natural	STAT3	PDAC cell lines	Silencing STAT3	[133]
N-acetylcysteine	Natural	JAK-STAT pathway	ICH model mice and rats	GSH increase; ROS generation inhibition; Lipid peroxidation inhibition	[75,76,77,78,79,80,81,82,83,84,85,86,87,88,89,90,91,92,93,94,95,96,97,98,99,100,101,102,103,104,105,106,107,108,109,110,111,112,113,114,115,116,117,118,119,120,121,122,123,124,125,126,127,128,129,130,131,132,133,134,135,136,137,138,139,140,141,142,143,144,145,146,147,148,149,150,151,152,153,154,155,156,157,158,159,160,161,162,163,164,165,166,167,168,169,170,171,172,173,174,175,176]
Curcumin	Natural	Iron	Mouse models; Mouse insulinoma pancreatic cells (MIN6)	GPx4 increased expression; Lipid peroxidation inhibition; Iron chelation	[179,180,181,182]
Epigallocatechin gallate (EGCG)	Natural	Iron	Mouse insulinoma pancreatic cells (MIN6)	Iron chelation; Lipid peroxidation inhibition	[181,182,183]
Apigenin	Natural	GPx4; SIRT1	Human neuroblastoma SH-SY5Y cells	ROS generation inhibition; GPx4 and SIRT1 induction	[184]
Quercetin	Natural	Iron	Mouse cells	Lipid peroxidation inhibition; Iron chelation	[185,186,187]
Baicalein	Natural	GPx4; Nrf2	Human pancreatic cancer cells (BxPc3); Epithelioid carcinoma cells (PANC1)	ROS generation inhibition; Lipid peroxidation inhibition	[188]

**Table 3 ijms-24-17279-t003:** Natural and synthetic inducers of ferroptosis.

Drugs	Nature of the Compound	Targets	Test Models	Functions	References
RSL-3	Synthetic	GPx4	Cancer cells	GPx4 inhibition; ROS generation inhibition	[137,138,139]
Erastin	Synthetic	System Xc^−^; VDAC2;	Cancer cells	VDAC2 inhibition; System Xc^−^ inhibition; GSH reduction	[140,141,142,143,144,145]
FIN56	Synthetic	GPx4	Cancer cells	GPx4 depletion; Squalene synthase activation	[146,147,148,149,150]
Artemisinin	Natural	Iron	Cancer cells	ROS generation Lipid peroxidation increase	[2,164,165,166]
Isothiocyanates (ITCs)	Natural	Iron MAPK signaling pathway	Cancer cells	ROS generation	[167,168,169]
Gambogic acid (GA)	Natural	Thioredoxin system	Cancer cells	ROS generation	[170,171,172,173]

## Data Availability

Not applicable.

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
