# Peer review of "Ferroptosis: Emerging Role in Diseases and Potential Implication of Bioactive Compounds"

_ijms, 2023, doi:10.3390/ijms242417279_

Round 1

Reviewer 1 Report (Previous Reviewer 3)

Comments and Suggestions for Authors

The presented review is devoted to ferroptosis, one of the ways of cell death, in which the main role is played by the excessive accumulation of free iron molecules. The authors faced a difficult task. Recently, many interesting and detailed reviews have appeared devoted to both the mechanisms of ferroptosis, its physiological functions, and its role in the pathogenesis of various diseases. A special place is occupied by the consideration of inducers and blockers of ferroptosis, in particular of natural origin, which can be of significant importance in the development of therapy. Such reviews have also been published in IJMS. For example,

1)         Yumin Wang et al.,Pharmacological Inhibition of Ferroptosis as a Therapeutic Target for Neurodegenerative Diseases and Strokes Int J Mol Sci. 2020 Nov; 21(22): 8765.

2)         Stockwell Ferroptosis turns 10: Emerging mechanisms,physiological functions,and therapeutic applications. Cell . 2022 Jul 7;185(14):2401-2421. doi: 10.1016/j.cell.2022.06.003

3)         Ge C et al. Emerging Mechanisms and Disease Implications of Ferroptosis: Potential Applications of Natural Products. Front Cell Dev Biol. 2022 Jan 18;9:774957. doi: 10.3389/fcell.2021.774957. eCollection 2021.

4)         Adrian Bartos, Joanna Sikora Bioinorganic Modulators of Ferroptosis: A Review of Recent Findings Int J Mol Sci . 2023 Feb 11;24(4):3634. doi: 10.3390/ijms24043634.

5)         Višnja Stepanić, Marta Kučerová-Chlupáčová Review and Chemoinformatic Analysis of Ferroptosis Modulators with a Focus on Natural Plant Products Molecules. 2023 Jan 4;28(2):475. doi: 10.3390/molecules28020475.

6)         Zhao et al.  Mechanisms of ferroptosis in Alzheimer's disease and therapeutic effects of natural plant products: A review. Biomed Pharmacother. 2023 Aug;164:114312. doi: 10.1016/j.biopha.2023.114312.

 In my opinion, the authors failed to cope with this task, perhaps due to the fact that they are not specialists in this field. It is not explained why these particular diseases are considered (cancer, neurodegenerative diseases, cardiovascular diseases), why from a large list of artificial and natural modulators of ferroptosis the authors chose only a few, describing their action and indicating them in the table. However, there is no reference to the fact that this list is much longer. It is necessary to finalize the conclusion, reflecting the authors’ personal opinion on the problem, to describe in more detail the prospects for the use of modulators. Now these are general conclusions that can be found in previously published reviews. It is also necessary to clearly indicate in the text of the manuscript what the novelty of this review is.

Comments on the Quality of English Language

 Minor editing of English language

Author Response

Dear Editor,

We would like to thank you the referees for their time spent on reviewing our manuscript (Manuscript ID: ijms-2713341) and for the comments helping us to improve the article. Their inputs and suggestions have been taken under consideration and they have been implemented in the manuscript. The changes within the text of the revised manuscript are clearly marked by red color, for easy identification. Below, we have tried to answer the questions and reply to the referee comments, point by point.

We hope that the changes have made the paper more appropriate for publication, and we look forward to your response.

Sincerely,

Ester Tellone

Reviewer comments:

Reviewer 1

The presented review is devoted to ferroptosis, one of the ways of cell death, in which the main role is played by the excessive accumulation of free iron molecules. The authors faced a difficult task. Recently, many interesting and detailed reviews have appeared devoted to both the mechanisms of ferroptosis, its physiological functions, and its role in the pathogenesis of various diseases. A special place is occupied by the consideration of inducers and blockers of ferroptosis, in particular of natural origin, which can be of significant importance in the development of therapy. Such reviews have also been published in IJMS. For example,

1) Yumin Wang et al., Pharmacological Inhibition of Ferroptosis as a Therapeutic Target for Neurodegenerative Diseases and Strokes Int J Mol Sci. 2020 Nov; 21(22): 8765.

2) Stockwell Ferroptosis turns 10: Emerging mechanisms, physiological functions, and therapeutic applications. Cell. 2022 Jul 7;185(14):2401-2421. doi: 10.1016/j.cell.2022.06.003

3)  Ge C et al. Emerging Mechanisms and Disease Implications of Ferroptosis: Potential Applications of Natural Products. Front Cell Dev Biol. 2022 Jan 18; 9:774957. doi: 10.3389/fcell.2021.774957. eCollection 2021.

4) Adrian Bartos, Joanna Sikora Bioinorganic Modulators of Ferroptosis: A Review of Recent Findings Int J Mol Sci. 2023 Feb 11;24(4):3634. doi: 10.3390/ijms24043634.

5) Višnja Stepanić, Marta Kučerová-Chlupáčová Review and Chemoinformatic Analysis of Ferroptosis Modulators with a Focus on Natural Plant Products Molecules. 2023 Jan 4;28(2):475. doi: 10.3390/molecules28020475.

6) Zhao et al.  Mechanisms of ferroptosis in Alzheimer's disease and therapeutic effects of natural plant products: A review. Biomed Pharmacother. 2023 Aug; 164:114312. doi: 10.1016/j.biopha.2023.114312.

In my opinion, the authors failed to cope with this task, perhaps due to the fact that they are not specialists in this field. It is not explained why these particular diseases are considered (cancer, neurodegenerative diseases, cardiovascular diseases), why from a large list of artificial and natural modulators of ferroptosis the authors chose only a few, describing their action and indicating them in the table. However, there is no reference to the fact that this list is much longer.

It is necessary to finalize the conclusion, reflecting the authors’ personal opinion on the problem, to describe in more detail the prospects for the use of modulators. Now these are general conclusions that can be found in previously published reviews. It is also necessary to clearly indicate in the text of the manuscript what the novelty of this review is.

According to the reviewer suggestion, the manuscript has been entirely revised and the conclusion section has been improved.

Reviewer 2 Report (Previous Reviewer 5)

Comments and Suggestions for Authors

The author has responded to the reviewer's concerns and has no further comment

Author Response

We thank the reviewer that helped us to improve the manuscript.

Reviewer 3 Report (New Reviewer)

Comments and Suggestions for Authors

This review summarizes the current knowledge on the mechanism of ferroptosis and ferroptosis inducers or inhibitors in the treatment of diseases.

Overall, the paper is clearly structured and well written. Nevertheless, there are some critical issues that should be addressed.

Major:

Line 287: From the figure2, it can be seen that ferroptosis is associated with many diseases. I don't understand why only three diseases (cardiovascular diseases, neurodegenerative diseases, and cancer) are discussed. I believe at least ischemia-reperfusion injury can be considered as a separate section.

Minor:

line 81:"ferroptosis induction can inhibit the proliferation of tumor cells."

Inducing ferroptosis may not be simply described as inhibiting proliferation. A better summary of the role of ferroptosis in cancer treatment is needed. Please refer to and cite the literature (PMID: 33514910) for more information.

line 102:"Ferrous iron through Fenton reaction increases ROS such as hydroxyl radical" may change to "Ferrous iron-mediated Fenton reaction increases ROS such as hydroxyl radical"

Line 116: "we find iron bound to the active site of numerous enzymes"

"we find" can be removed

Line 313: "Therefore, a reduction in targets of Nrf2 (an upstream transcriptional regulator of SLC7A11 and GPx4), such as GPx4,"

"such as GPx4" can be removed.

Line 317: "the reduced expression of SLC7A11 gene which encodes Xc- system" the discription is not accurate. The system Xc- includes SLC7A11 and SLC3A2.

Line 331: "Gamma interferon" may change to "interferon gamma (IFN-γ)"

Line 471: "RSL-3) is a compound that selectively inhibits GPx4" may change to "RSL-3) is a compound that selectively inhibits the activity of GPx4"

Line 476-485: I suggest to delete the discription about eratin's inhibition on VDAC. It is now widely believed that erastin inhibits SLC7A11.

Line 491-505: The description of FIN56 needs to be simplified. In addition, other GPX4 degrader should be shortly discripted such as Bufotalin (PMID: 35038550 ) and PdPT (PMID 32596160). 

Line 520:  "Similarly, to ferrostatin-1, Liproxstatin-1 (Lipro-1) can block ferroptosis even in the presence of known ferroptosis inducers such as FIN56 or RSL-3"

"even" should be removed.

Author Response

Line 287: From the figure2, it can be seen that ferroptosis is associated with many diseases. I don't understand why only three diseases (cardiovascular diseases, neurodegenerative diseases, and cancer) are discussed. I believe at least ischemia-reperfusion injury can be considered as a separate section.

As suggested ischemia-reperfusion injury has been treated in a separate section.

Minor:

line 81:"ferroptosis induction can inhibit the proliferation of tumor cells."

Inducing ferroptosis may not be simply described as inhibiting proliferation. A better summary of the role of ferroptosis in cancer treatment is needed. Please refer to and cite the literature (PMID: 33514910) for more information.

The reference suggested helped us to improve the role of ferroptosis in cancer. The literature has been cited and added.

line 102:"Ferrous iron through Fenton reaction increases ROS such as hydroxyl radical" may change to "Ferrous iron-mediated Fenton reaction increases ROS such as hydroxyl radical”.

The sentence has been corrected.

Line 116: "we find iron bound to the active site of numerous enzymes”.

"we find" can be removed

"we find" has been removed.

Line 313: "Therefore, a reduction in targets of Nrf2 (an upstream transcriptional regulator of SLC7A11 and GPx4), such as GPx4,"

"such as GPx4" can be removed.

"such as GPx4" has been removed.

Line 317: "the reduced expression of SLC7A11 gene which encodes Xc- system" the discription is not accurate. The system Xc- includes SLC7A11 and SLC3A2.

The sentence has been corrected.

Line 331: "Gamma interferon" may change to "interferon gamma (IFN-γ)"

The sentence has been corrected.

Line 471: "RSL-3) is a compound that selectively inhibits GPx4" may change to "RSL-3) is a compound that selectively inhibits the activity of GPx4".

The sentence has been corrected.

Line 476-485: I suggest to delete the discription about eratin's inhibition on VDAC. It is now widely believed that erastin inhibits SLC7A11.

As suggested the description has been deleted.

Line 491-505: The description of FIN56 needs to be simplified. In addition, other GPX4 degrader should be shortly discripted such as Bufotalin (PMID: 35038550 ) and PdPT (PMID 32596160).

According with suggestion, the description has been simplified, other GPx4 degraders have been descripted and literature has been cited.

Line 520:  "Similarly, to ferrostatin-1, Liproxstatin-1 (Lipro-1) can block ferroptosis even in the presence of known ferroptosis inducers such as FIN56 or RSL-3".

"even" should be removed.

The sentence has been corrected.

Reviewer 4 Report (New Reviewer)

Comments and Suggestions for Authors

In this review manuscript, Patanè et al, aims to provide a comprehensive understanding of the mechanisms underlying ferroptosis and the latest advancements in its regulatory pathways. The authors also discuss how ferroptosis is linked to different diseases and highlights its potential as a target for pharmacological treatment. The manuscript is thorough and well written, however, there are certain concerns that needs to be addressed.

1. In the introduction, the authors state: Interestingly, oxidative stress and lipid peroxidation, two of the main characteristics of ferroptosis are main features also of inflammation diseases and compelling evidence demonstrates inflammatory cytokines, such as tumor necrosis factor alpha (TNF-α) and interleukine-6 (IL-6), can induce ferroptosis in cancer cells [2, 5,6]. What inflammation diseases are being referred to? Please provide appropriate references.

2. In the introduction, the authors talk about PD and ferroptosis. Ferroptosis in the context of other neurodegenerative diseases (AD, HD and ALS) should also be discussed briefly.

3. The authors talk about the application of ferroptosis regulators in cancer therapy. The authors should also include the recent advances of ferroptosisis modulators in the context of neurodegenerative disorders.

4. For Tables 2 and 3, it will be informative to indicate in the table which inhibitors are natural and the ones that are synthetic. 

6. Expand all the abbreviations used in the text. Some are missing. For example RAS, COPZ1 etc.

Comments on the Quality of English Language

There are a lot of typos and grammatical errors that should be fixed. For example:

1. Lines 130-132: Into the blood, iron circulates bound to transferrin (Tf), a glycoprotein consisting of a single polypeptide chain that possesses two binding sites for the ferric ion (Fe3+).

2. Lines 248-249: Erythroid nuclear factor 2-related factor (Nrf2) is an antioxidant agent which action may contribute to the onset of ferroptosis.

Author Response

  1. In the introduction, the authors state: Interestingly, oxidative stress and lipid peroxidation, two of the main characteristics of ferroptosis are main features also of inflammation diseases and compelling evidence demonstrates inflammatory cytokines, such as tumor necrosis factor alpha (TNF-α) and interleukine-6 (IL-6), can induce ferroptosis in cancer cells [2, 5,6]. What inflammation diseases are being referred to? Please provide appropriate references.

References have been corrected.

  1. In the introduction, the authors talk about PD and ferroptosis. Ferroptosis in the context of other neurodegenerative diseases (AD, HD and ALS) should also be discussed briefly.

As suggested the introduction has been implemented.

  1. The authors talk about the application of ferroptosis regulators in cancer therapy. The authors should also include the recent advances of ferroptosisis modulators in the context of neurodegenerative disorders.

According to the reviewer suggestion this section has been improved.

  1. For Tables 2 and 3, it will be informative to indicate in the table which inhibitors are natural and the ones that are synthetic. 

As suggested tables have been supplemented.

  1. Expand all the abbreviations used in the text. Some are missing. For example RAS, COPZ1 etc.

All the abbreviations have been expanded.

Comments on the Quality of English Language

There are a lot of typos and grammatical errors that should be fixed. For example:

  1. Lines 130-132: Into the blood, iron circulates bound to transferrin (Tf), a glycoprotein consisting of a single polypeptide chain that possesses two binding sites for the ferric ion (Fe3+).

The sentence has been corrected.

  1. Lines 248-249: Erythroid nuclear factor 2-related factor (Nrf2) is an antioxidant agent which action may contribute to the onset of ferroptosis.

The sentence has been corrected.

Round 2

Reviewer 1 Report (Previous Reviewer 3)

Comments and Suggestions for Authors

The manuscript was improved and can be published in IJMS.

Comments on the Quality of English Language

Minor editing of English language

Reviewer 3 Report (New Reviewer)

Comments and Suggestions for Authors

The authors have addressed all my concerns.

This manuscript is a resubmission of an earlier submission. The following is a list of the peer review reports and author responses from that submission.

Round 1

Reviewer 1 Report

Comments and Suggestions for Authors

The present review outlined the current progress on the mechanisms and regulatory pathways of ferroptosis, and its involvement in the diseases. Subsequently, several types of ferroptosis inducers and inhibitors, especially from naturally occurring phytochemicals, were discussed to highlight new perspective on potentially pharmacological clinical therapies against ferroptosis-related diseases.

GSH-GPx4 pathway was emphasized, but some newly-identified GSH-GPx4-independent pathways, such as FSP1, dihydroorotate dehydrogenase, etc. were skimmed over, limiting the novelty. These pathways should be combined into Figure 1 and summarized respectively and then discussed their potential links as a separate section.

The current knowledge involving ferroptosis was summarized but the potential issues and some contradictory findings were hardly discussed.

Future direction as a separate section should be supplemented.

Comments on the Quality of English Language

Overall, the English language is qualified, in spite of a few inadequacy of the expression

Reviewer 2 Report

Comments and Suggestions for Authors

The authors listed and adequately described all relevant scientific facts about ferroptosis and its relations with selected diseases. Moreover, the effects of natural compounds on ferroptosis have also been reviewed in the manuscript.

The addition of several figures could improve the overall quality of the manuscript making it easier for readers to follow, by my opinion. 

Although the relevant literature is citated, the methodology for literature research could be added. 

Reviewer 3 Report

Comments and Suggestions for Authors

The reviewed work is devoted to one of the types of cell death - ferroptosis. Possible molecular mechanisms of this process, its role in the development of diseases and the possibility of treating these diseases by influencing this process are considered. The topic of the review is interesting and relevant. However, the presented review does not contain new information, but in fact is a very reduced and weak version of the previously published work by Ge C, Zhang S, Mu H, Zheng S, Tan Z, Huang X, Xu C, Zou J, Zhu Y, Feng D, Aa J. Emerging Mechanisms and Disease Implications of Ferroptosis: Potential Applications of Natural Products. Front Cell Dev Biol. 2022 Jan 18;9:774957. doi: 10.3389/fcell.2021.774957.  Also, figure 2 almost coincides with figure 3 from Ge et al. It should be noted that, unlike Ge et al., Patanè et al. considers the role of ferroptosis in the development of neurodegenerative diseases in more detail in the text of the article, while Ge et al. present a lot of information in the table.

Minor remarks.

1.        Tables 1 and 2 have the same information.

2        The text contains typos.

3         Also in the reviewed work there are naive statements, such as “The heart is one of the most important organs in the body. Its main function is to pump oxygen-rich blood to all parts of the body….” (lines 403-404).

The authors of the manuscript under review have done some work, and I hope that the material presented will be used by them to write a truly original review.

Comments on the Quality of English Language

Minor editing of English language required

Reviewer 4 Report

Comments and Suggestions for Authors

In this review article, the authors tries to summarize the current knowledge on the mechanism of ferroptosis and the latest advances in its multiple regulatory pathways, underlining ferroptosis involvement in the diseases. However, the manuscript did not achieve this goal. The authors only introduce the knowledge on the mechanism of ferroptosis several years ago. Few reference of part 2 and 3 are published after 2020. Many new findings on the mechanism of ferroptosis in recent years are not mentioned. Most importantly, it is well accepted nowadays that ferroptosis is not only regulated by system Xc– /GSH/GPX4, but also regulated by many antioxidant pathways, such as FSP1/CoQ10, GCH1/BH4, DHODH/ CoQ10, etc. The authors missed a huge part of the mechanism of ferroptosis. They even put FSP1 in the part 5. It is also not well summarized the diseases related to ferroptosis. In addition to cancer, neurological diseases, and cardiovascular diseases, ferroptosis is also involved in the injuries of many other organs, such as kidney, lung, and liver, as well as some special diseases (e.g. preeclampsia).

The title of this review article is “Ferroptosis emerging role in diseases and powerful effects of natural compounds”. The authors also stated that they will focuse on several types of ferroptosis inducers and inhibitors in this review. However, they talk about many synthesized compounds, the natural compounds are only a small part. And many recent finds are also not included. For example, many FSP1 inhibitors have been reported, but the authors only state “Future studies with the aim to synthesize molecules acting as FSP1inhibitors will may improve cancer therapy and treatment especially for those tumor forms highly resistant to pharmacological treatments.” Moreover, the authors make many mistakes in the Table 3. FSP1 is not a synthetic modulator or a natural modulator for ferroptosis. It is a protein like GPX4. The target of Ferrostatin-1, lipro-1, and N-acetylcysteine are totally wrong.

As there is too many obvious mistakes in this review article, I am not sure the understanding of authors on ferroptosis is right.

Reviewer 5 Report

Comments and Suggestions for Authors

In this review manuscript, the authors begin by briefly describing an overview of ferroptosis. The authors then also provide a relevant summary of the mechanisms of ferroptosis, as well as an account of the relationship between ferroptosis and disease. Finally, the authors also summarize the pharmacological role of natural compounds as inducers or inhibitors of ferroptosis in the treatment of disease. Following are some minor comments:

1. Table 1 and Table 2 look too simple and the contents are too pedestrian.

2. In line 55, the discoverer of the ferroptosis is not "Dixon" but "Dr Brent R Stockwell".

3. It is noted that your manuscript needs careful editing by someone with expertise in technical English editing paying particular attention to English grammar, spelling, and sentence structure so that the goals and results of the study are clear to the reader. For example, in line 92, the word "mechanism" is misspelled.

4. In part 5, "Pharmacological approaches to ferroptosis modulation", and part 6, "Natural Ferroptosis inducers and inhibitors", the articles are too long and redundant, making them very difficult to read, so it is recommended that the narrative be divided into paragraphs and be concise and clear.

5. References: There were some minor issues with the references, including incomplete citations or outdated sources. Please check the references carefully and make the necessary corrections. For example, [22], [38], [46], [55], [56], [58], [59], [103], [117].

6. In this manuscript, the potential value of the research is unclear. We recommend that you add the Highlight section to your manuscript so that you can explain this aspect to your readers more clearly and convincingly.

Comments on the Quality of English Language

In this review manuscript, the authors begin by briefly describing an overview of ferroptosis. The authors then also provide a relevant summary of the mechanisms of ferroptosis, as well as an account of the relationship between ferroptosis and disease. Finally, the authors also summarize the pharmacological role of natural compounds as inducers or inhibitors of ferroptosis in the treatment of disease. Following are some minor comments:

1. Table 1 and Table 2 look too simple and the contents are too pedestrian.

2. In line 55, the discoverer of the ferroptosis is not "Dixon" but "Dr Brent R Stockwell".

3. It is noted that your manuscript needs careful editing by someone with expertise in technical English editing paying particular attention to English grammar, spelling, and sentence structure so that the goals and results of the study are clear to the reader. For example, in line 92, the word "mechanism" is misspelled.

4. In part 5, "Pharmacological approaches to ferroptosis modulation", and part 6, "Natural Ferroptosis inducers and inhibitors", the articles are too long and redundant, making them very difficult to read, so it is recommended that the narrative be divided into paragraphs and be concise and clear.

5. References: There were some minor issues with the references, including incomplete citations or outdated sources. Please check the references carefully and make the necessary corrections. For example, [22], [38], [46], [55], [56], [58], [59], [103], [117].

6. In this manuscript, the potential value of the research is unclear. We recommend that you add the Highlight section to your manuscript so that you can explain this aspect to your readers more clearly and convincingly.